# Diet and Non-Alcoholic Fatty Liver Disease: The Mediterranean Way

**DOI:** 10.3390/ijerph16173011

**Published:** 2019-08-21

**Authors:** Ludovico Abenavoli, Luigi Boccuto, Alessandro Federico, Marcello Dallio, Carmelina Loguercio, Laura Di Renzo, Antonino De Lorenzo

**Affiliations:** 1Department of Health Sciences, University “Magna Graecia”, Viale Europa—Germaneto, 88110 Catanzaro, Italy; 2Greenwood Genetic Center, 113 Gregor Mendel Circle, Greenwood, SC 29646, USA; 3Department of Precision Medicine, University of Campania “Luigi Vanvitelli”, via Pansini 5, 80131 Naples, Italy; 4Department of Biomedicine and Prevention, Section of Clinical Nutrition and Nutrigenomic, University of Rome Tor Vergata, Via Montpellier 1, 00133 Rome, Italy

**Keywords:** metabolism, steatosis, inflammation, polyphenols, microbiota

## Abstract

Lifestyle interventions remain the first-line treatment for non-alcoholic fatty liver disease (NAFLD), even if the optimal alimentary regimen is still controversial. The interest in antioxidants has increased over time, and literature reports an inverse association between nutrients rich in antioxidants and the risk of mortality due to non-communicable diseases, including NAFLD. Mediterranean diet (MD) is a model characterized by main consumption of plant-based foods and fish and reduced consumption of meat and dairy products. MD represents the gold standard in preventive medicine, probably due to the harmonic combination of many foods with antioxidant and anti-inflammatory properties. This regimen contributes substantially to the reduction of the onset of many chronic diseases as cardiovascular diseases, hypertension, type 2 diabetes mellitus, obesity, cancer, and NAFLD. The present review aims to clarify the intake of antioxidants typical of the MD and evaluate their effect on NAFLD.

## 1. Introduction

Non-alcoholic fatty liver disease (NAFLD), is the hepatic manifestation of the metabolic syndrome associated with obesity [1]. Metabolic changes, including insulin resistance and impaired lipid metabolism, have been hypothesized to contribute to molecular pathogenesis of NAFLD [2,3]. Moreover, steatosis is a necessary condition but not a sufficient component, for the development of nonalcoholic steatohepatitis (NASH) [4]. Steatosis is related to the imbalance in triglyceride metabolism, including uptake, clearance and removal [5]. Literature indicate that the severity of hepatic fat accumulation predicts the risk of steatohepatitis, as well as the risk of progression to cirrhosis [6]. Additionally, high body mass index, is a pivotal risk factor for steatosis, suggesting that excess caloric intake and obesity contribute to NAFLD development [1,7]. In addition, genetic polymorphisms, and in particular the p.I148M polymorphism of the Patatin-like phospholipase domain-containing protein 3 (PNPLA3) gene, have also been shown to increase the risk of the onset for NAFLD/NASH [8]. Furthermore, the gut is involved in the pathogenesis of NAFLD through metabolism of nutrients, hormones, toxins and microbiota disequilibrium that targets the liver, and forms the so-called “gut-liver axis” [9].

The overall incidence of NAFLD has been growing rapidly, and consequently this disorder has emerged as a leading cause of chronic liver disease worldwide [10]. NAFLD is also an increasingly cause of liver cirrhosis and is expected to soon become the main cause of liver transplantation in Western Countries. NAFLD affects 30% of the adult population worldwide, and the majority of obese people, which makes obesity the principal driver of this disease [10]. Fatty liver results primarily form high calorie intake, and lack of physical activity in a genetic predisposing context [11]. Therefore, considering how NAFLD is the main liver disease influenced by dietary profile, lifestyle changes have become pivotal in the clinical approach to this disorder described in current guidelines [12].

The burden of NAFLD has been not only restricted to the liver, but it has been known that NAFLD is a multisystem disease with extra-hepatic manifestations. NAFLD, in fact, increases the risk of type 2 diabetes mellitus, cardiovascular and cardiac diseases, and chronic kidney disease [1,13].

Traditional Mediterranean diet (MD) is an alimentary model characterized by high consumption of whole cereals, fruit, legumes, vegetables, and nuts, a moderate use of dairy products, a low consumption of meat and poultry and a moderate consumption of alcohol, i.e., red wine during the meals [14,15,16]. The consumption of olive oil is high while the saturated fatty acids intake is also low. The Mediterranean regimen is closely connected to the cultural practices. The United Nations Educational, Scientific and Cultural Organization (UNESCO) listed it in 2010 under the heading MD, for this reason, on the Representative List of the Intangible Cultural Heritage of Humanity [17].

The health properties and beneficial effects of MD in patients with NAFLD, have been described for liver steatosis and metabolic dysfunction, based on observational studies and short-term trials with consistent results [16]. This narrative review aims to describe current acknowledgments and to highlight new perspectives, on the use of MD in patients with NAFLD and its related metabolic features.

## 2. Mediterranean Diet and Antioxidant Foods

The interest in antioxidants and polyphenols has increased over time, as epidemiology has indicated an inverse association between the presence of nutrients rich in antioxidants and the risk of mortality from non-communicable diseases [18]. For decades, the antioxidant compounds present in some nutrients have been considered only powerful “scavengers of free radicals”. Then, their action has been related to several biological effects such as anti-inflammatory action, inhibition of tumor proliferation, cholesterol absorption, and modulation of different enzymes including telomerase and others implicated in redox reactions [19]. In particular, in carcinogenesis there are potential chemo-preventive mechanisms such as the modulation of the energy metabolism of a carcinoma, pathway regulation, inhibition of cell proliferation and apoptotic induction capacity [19,20].

Furthermore, in vitro, in vivo, and epidemiological studies have shown the ability of these bioactive molecules to positively influence the pathogenic steps of atherogenesis, including oxidation of low-density lipoproteins, release of nitric oxide, inflammation, oxidative stress, chemotaxis, cell adhesion, foam cell formation, smooth muscle cell proliferation, and platelet aggregation [18,21].

However, a powerful antioxidant activity detected in vitro may be absent in humans due to poor bioavailability. A potent bioactive compound, even if administered in large quantities, is not able to exert its beneficial effects in the target tissues if it does not reach its site of action [22]. Therefore, for the prevention of diseases and the improvement of human health it is necessary to know the biological capacities, the bioavailability and the metabolites of the polyphenols contained in foods to choose the best dietary profile [23]. In MD, foods mainly rich in polyphenols are fruits, vegetables, red wine, wild herbs, spices, nuts, and also olives and especially extra virgin olive oil. Not least, today, influences from the heritage of other cultures have enriched our diet with other elements in this pattern with high concentration of polyphenols such as turmeric, spices, coffee, green tea, and chocolate [24,25].

In general, mixtures of polyphenols with different concentrations in the same category are contained in different foods. For example, in red wine there are flavonols, flavanols, proanthocyanidins, anthocyanins, phenolic acids, hydroxycinnamates and stilbenes, in particular resveratrol [26]. Even in extra virgin olive oil there is a great variability in concentration and composition, in fact there are up to 36 phenolic compounds and the total concentration varies between 0.02–600 mg/kg [27]. The intake of antioxidants, in particular of polyphenols, has been correlated with a reduction in mortality rates [28]. A systematic review on the use of antioxidant supplementation in healthy individuals and subjects affected by non-communicable diseases has failed to find an association with a reduction in mortality assessed for all causes. Indeed, in the analysis of single individual studies it has been found that vitamin antioxidants were associated with higher mortality from all causes [29]. The causes of this phenomenon are due to the reduced intake of micronutrients and the reduced absorption of these in an isolated form compared to the phytocompounds. In this way, it has been highlighted that the adequacy of food choices to the Mediterranean pattern is an essential objective to maintain the state of health [30].

The role and effectiveness of the MD, supplemented with extra virgin olive oil, nuts, red wine, in the prevention of cardiovascular events in high risk subjects have been recently confirmed [31]. The Mediterranean pattern has been proven beneficious in epidemiological studies, probably due to the strength of the phytocompounds and its modulation of the microbiome, capable of reducing the onset of non-communicable diseases, their complications and mortality [32]. Furthermore, the presence of antioxidants is a high qualitative and nutritional standard index and in the choice of food the origin, conventional or organic, of fruit and vegetables must be taken into account. Therefore, it is necessary to prefer vegetables of organic origin since in the absence of pesticides or insecticides it produces antioxidant bioactive molecules useful for growth in a competitive environment [33].

Adding a quota of antioxidants in the diet can prevent the metabolic syndrome and its manifestations including obesity and NAFLD [34]. Contextually, a controlled food chain is necessary in order to ensure the absence of contaminants and suitable antioxidants in the diet. Otherwise, the risk is to consume sub-optimal, harmful and poor food for human health. The Nutrient Hazard Analysis and Critical Control Point (NACCP) process represents the correct way of monitoring the nutritional quality of foods of the selected dietary pattern, in order to have the highest antioxidant content. In fact, it is possible to monitor the entire production chain and identify the critical points where to intervening to improve the quality of nutrients [35]. Special attention must be paid to the measurement of standards for antioxidants quality and quantity. However, it remains clear how adherence to the MD increases quality and life expectancy by reducing the risk of death, also related to higher content of antioxidants and polyphenols mainly present in foods rich in fiber. It leads to a modulation of the gut microbiota that influences, and it is influenced by, the polyphenols and the foods that contain them [36,37]. In particular, the intestinal microflora is able to metabolize and make certain substances absorbable. Furthermore, a phytocompound is superior to the corresponding isolated compound in terms of yield on the absorption and action of polyphenols and antioxidants. This phenomenon is attributable to the components present in the food that may increase in synergy the bioavailability or the metabolic action of probiotics [38].

## 3. Mediterranean Diet and Liver Steatosis

Currently, NAFLD has become an emerging public health problem worldwide, due to its increasing prevalence [5]. Efforts to diagnosis, prevention and treatment are mandatory and must be so in the future, since the NALFD can progress in cirrhosis and its complications including cancer and is related to cardiovascular and metabolic diseases [39,40]. Many of the genetic factors predisposing to NAFLD suggest a critical role for the lipid metabolism and inflammation, which ultimately affect intracellular oxidative processes [41,42]. Examples are the p.I148M polymorphism in the PNPLA3 gene promoting lipid accumulation in hepatocytes, or Sirtuin abnormalities affecting β-oxidation of fat acids and lipogenesis, or disruption of the Nrf2 signalling pathway that have repercussions on the expression of genes encoding proteins with antioxidant activity and on the inflammation-regulating Nf-κB signalling pathway [42]. Based on the risk factors underlying the development of NAFLD, the treatment that has showed the best result is the multidisciplinary approach, with a team including dietitian, psychologist, and physical activity trainer and it is the suitable method in the management of NAFLD patients [43]. The nutritional intervention that drives the adhesion to a Mediterranean dietary pattern and physical activity is more effective than any single pharmacological option [1]. Treatment for NAFLD involves a constant change in lifestyle habits. Literature data and international guidelines have highlighted the health benefits of weight loss and exercise [12,44]. Physical activity, suggested in 200–300 min/week, is essential to achieve and maintain weight loss, with an independent positive effect on NAFLD treatment [45].

In this way, the MD seems to be the ideal diet for patients with NAFLD, thanks to its effectiveness on the liver status that leads to the improvement of insulin sensitivity and lipid profile, but also to be a primary form of prevention for metabolic related diseases [16,46]. Actually, the lack of a standardized pharmacological approach to NAFLD/NASH management, focuses the treatment on associated/co-existing diseases, as a diabetes, obesity, and lipid disorders, to control the liver function, glycemic and lipidic profile [47,48]. Based on the available literature, it is possible to summarize the following dietary recommendations for patients with NAFLD: (1) expected reduction of the body weight within 6 months 5–7% in NAFLD, 7–10% in NASH; (2) expected rate of weight loss 0.5–1 kg/week, 1.5 kg/week in severe obesity; (3) caloricity of the diet: women: 1200–1500 kcal/day for women, 1500–1800 kcal/day for men [12,49]. The adherence of MD in patients with NAFLD results in changes of clinical and biochemical parameters. In particular, MD not only improve clinical parameters as a weight, waist circumference, hepatic fat accumulation, the blood levels of transaminases, gamma-glutamyltransferase, triglycerides, cholesterol, insulin and insulin-resistance, but also it positively influences inflammatory biomarkers such as adhesion molecules, cytokines or molecules related to the stability of atheromatic plaque [50].

Indeed, the combined effect of MD and physical activity approach is able to improve not only anthropometric parameters and biochemical profile, but also hepatic fat accumulation [51,52]. Based on the pathogenic model based on free radicals production, antioxidant supplements were selected to be administered in combination with MD and such combined strategy led to amelioration of insulin resistance [52]. It is also important to note that the association of probiotics with MD and physical activity presents a synergistic action in maintaining liver health. Among the many species of bacteria, *Lactobacillus* seems to be the most promising in the treatment of NAFLD [53,54]. In fact, probiotics supplementation is able to modulate the gut microbiota and to perform health benefits to the host. The probiotics themselves, their vitality, their balance and their action must be supported through prebiotics, plant fibers and polyphenols: all molecules typically present in MD [55,56].

Over time, research has shifted the attention from the single nutrients to the alimentary pattern and defined as reference the post-war food model of Nicotera [57,58,59]. To evaluate and compare this food model, the Mediterranean adequacy index (MAI) has been developed. It is obtained from the ratio of the combined percentage of total energy from Mediterranean foods and the total energy from non-Mediterranean foods [60]. In particular, a daily score 5 is considered necessary to obtain a health benefit, while a meal that has a MAI 7 reduces postprandial inflammation, oxidative stress and positively modulates the expression of inflammatory genes [61]. In general, it was observed that populations with a high MAI food pattern had a low mortality risk. The reduction in mortality, inflammation and oxidative stress is attributed to the bioactivity of fibers, monounsaturated fatty acids, in particular ω-3, vitamin A, C and E, and polyphenols [62]. However, longitudinal studies on eating habits have shown a reduction of MAI in Italian regions [60]. To explain the MD, we use the template consisting of a balanced combination of fruit, vegetables, fish, legumes, cereals, and polyunsaturated fats from extra virgin olive oil, with a reduced consumption of meat and dairy products and a moderate consumption of alcohol [57]. The carbohydrates should be complex and starchy, from unrefined cereals, bread and pasta, legumes and vegetables; only 5% of carbohydrates must be from fruit or red wine. The energy deriving from ethyl alcohol, mainly the wine consumed during meals, can be included in acceptable quantities with 2 glasses a day for men and 1 for women. Vegetables are important elements of the MD as main source of phytosterols that reduce cholesterol and cardiovascular risk. The general recommendations for the diet must be personalized and depending on the body weight.

## 4. Mediterranean Diet in the Metabolic Context

In healthy subjects, several trials have shown the efficacy of antioxidant foods/drinks on gene expression, metabolic and inflammatory pathways, suggesting that the antioxidant status reflects the antioxidant content of foods [24,26]. Several diseases show altered pathways of oxidative stress, including cardiovascular diseases, cancer, neurodegenerative diseases, cataracts, macular degeneration [63]. For this reason, it is necessary to identify therapeutic strategies that contemplate the use of antioxidants formulation.

As already reported, fruits and vegetables are the main sources of these molecules, in particular of low molecular weight antioxidants that result particularly efficient at protecting the cells from oxidative stress. Alpha-tocopherol, ascorbic acid, beta-carotene would seem to have an effect on the reduction of the risk of many diseases, probably due not to a single antioxidant action but to a cumulative effect [64]

With its high intake of antioxidants, the MD contributes substantially to the reduction of cardiovascular risk and in particular to the reduction of incidence of thrombosis, hypertension, type 2 diabetes mellitus, and obesity, as demonstrated by the Primary Prevention of Cardiovascular Diseases with a Mediterranean Diet (PREDIMED) study [65]. On the other hand, it has been shown that oxidative stress predisposes to neurodegenerative diseases by affecting cellular function and structure. Thanks to the antioxidants taken with the diet, in particular the ones belonging to the polysaccharide class from vegetables, fruits, cereals, legumes, tea, nuts, mushrooms, probiotics, it is possible to ameliorate the signs and symptoms of neurodegenerative diseases, reducing cognitive and motor decline, intervening on mitochondrial function, antioxidant pathway and protein misfolding, as well as acting as scavengers against free radicals. In this case, the polysaccharides seem to show a double effect of oxidative stress reduction and protection against the related diseases [34,66,67].

Antioxidants can exert a positive effect on the obesity phenotype spectrum as defined by specific metabolic characteristics by limiting the inflammatory and oxidative state of the subjects, reducing the progression of cardiometabolic co-morbidities [68]. Several studies have also suggested a positive role of antioxidants in the treatment of cancer, but it is always necessary to pay attention to the absolute data [69]. For example, in the case of polyphenols, their poor bioavailability has highlighted the need for further studies to better understand the mechanisms of action in different tissues, as well as the influence of genetic variability [70]. Moreover, a low bioavailability limits the possibility of investigating the effective ability to modulate pathways regulating cell growth.

Finally, it is important to highlight that the use of drugs should be indicated for NASH histologically characterized by bridging fibrosis and/or necro-inflammatory activity, and for early-stage of steatohepatitis with risk factors for fibrosis progression, at an age 50 years, diabetes, and elevated transaminases blood levels [12]. Therefore, no specific drugs are actually recommended, and any treatment would be off-label. These recommendations elucidated by international guidelines, support and emphasize the therapeutic role of Mediterranean regimen in NAFLD patients [49].

## 5. Conclusions

Actually, a standardized treatment for NAFLD/NASH remains to be established, despite the consistent number of clinical trials that have been conducted. Considered the worldwide prevalence of NAFLD, the definition of a therapeutic algorithm, is the challenge in this field of the near future.

The influence of the diet in the pathogenesis of NAFLD has been thoroughly demonstrated. Nowadays, the cornerstone in the clinical management of NAFLD includes two easy and pivotal concepts, that are dietary modification associated with improvement of physical activity. In the era of evidence-based medicine, we conclude that MD can be considered the gold standard in preventive medicine targeting a vast spectrum of disorders involving the imbalance of the oxidative metabolism. Its benefits are due to the combination of several foods with antioxidant and anti-inflammatory properties, which overwhelm any single nutritive or alimentary element. On the basis of the data currently available in the literature, we suggest the prescription of MD in NAFLD patients, as an appropriate therapeutic approach, to prevent its onset and to contrast the development of severe forms. However, longer-term trials testing the health benefits effects of MD are warranted.

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
