# Peer review of "Diet and Non-Alcoholic Fatty Liver Disease: The Mediterranean Way"

_ijerph, 2019, doi:10.3390/ijerph16173011_

Round 1

Reviewer 1 Report

- In the Introduction Section. The authors showed that NAFLD is the hepatic manifestation of the metabolic syndrome associated with obesity; however, it may be relevant focused on the relations between NAFLD and metabolic pathway. Please consider these references (PMID:31010049, PMID:27237577, PMID:24656139, PMID:31336392).

- It may be interesting analyze the clinical aspects of mediterranean diet and treatments of metabolic factors associated with NAFLD, as glucose and lipid lowering treatments. In particular, it may be interesting to evaluate when mediterranean diet is possible and helpful wiht or without drugs or when the diet is not the only solution. Please see these references (PMID:29260404

PMID:30817261, PMID:31050706)

Author Response

Referee #1

Dear Referee,

thank you for your comments finalized to improve the submitted paper. The changes are marked in red in the text.

In particular:

- introduction section has been revised and the metabolic pathway has been reported

- the role of Mediterranean diet in the treatment of metabolic features, has been highlighted and new references are now reported

Reviewer 2 Report

Title: The title is not consistent with the subject matter.

General recommendation of the article:

I advise the authors of this review to conduct a search in PubMed or Web of Science on the published reviews on this subject, with the following strategy “non-alcoholic fatty liver OR non-alcoholic steatohepatitis AND Mediterranean diet”. The most interesting results obtained are:

Aller R, Fernández-Rodríguez C, Lo Iacono O, Bañares R, Abad J, Carrión JA, et al. Consensus document. Management of non-alcoholic fatty liver disease (NAFLD). Clinical practice guideline. Gastroenterol Hepatol. 2018;41(5):328-349. doi: 10.1016/j.gastrohep.2017.12.003. Recommended to review figure 1.

Effect of Mediterranean Diet and Antioxidant Formulation in Non-Alcoholic Fatty Liver Disease: A Randomized Study. Nutrients. 2017;9(8). pii: E870. doi: 10.3390/nu9080870. Recommended to review table 1 and 2. An example of the variation of markers.

Leoni S, Tovoli F, Napoli L, Serio I, Ferri S, Bolondi L. Current guidelines for the management of non-alcoholic fatty liver disease: A systematic review with comparative analysis. World J Gastroenterol. 2018;24(30):3361-3373. doi: 10.3748/wjg.v24.i30.3361. Recommended to review table 4.

Suárez M, Boqué N, Del Bas JM, Mayneris-Perxachs J, Arola L, Caimari A. Mediterranean Diet and Multi-Ingredient-Based Interventions for the Management of Non-Alcoholic Fatty Liver Disease. Nutrients. 2017;9(10). pii: E1052. doi: 10.3390/nu9101052. Recommended to review table 1.

The narrative development of this review should be improved in order, concepts that are repeated in the text, such as the concept of the Mediterranean diet and the prevalence of fatty liver disease are shown. The authors should direct the article in a specific thematic area if they indicate it, if they talk about lifestyle they will have to talk about all the possible options to treat this disease and if they talk about Mediterranean diet and exercise they should focus on this area. In my personal opinion, I would advise them to focus on the effect of the Mediterranean diet, physical exercise and non-alcoholic fatty liver disease.

I have sent you three articles that have been published recently about the Mediterranean diet and also about physical activity. These articles should be used as an example of how a review should show when an intervention is effective or not, indicating in which parameter or marker it affects (biochemical marker, liver fat ...).

With the analysis transmitted the authors will realize that the format of the article must be changed and that said change present a more direct, innovative and objective perspective.

Summary: It must be identified that it is a narrative review, not a scoping review, or a systematic review.

Article development:

This is a review article or indicate revision directly, this must be indicated in the article header, it is not an original article.

In the introduction:

The definition of non-alcoholic fatty liver disease should be improved and indicate one or more terms or abbreviations that are currently being used, such as non-alcoholic steatohepatitis (NASH) (Povsic M et al. 2019 from the bibliography of this article). Although finally in the article you decide to use an abbreviation and a specific term.

In the introduction you mention a poor Mediterranean diet definition and then complete it in the third section, it doesn't make much sense and it sounds repetitive.

The article presents approaches that I do not see related to the effectiveness of the Mediterranean diet, physical activity and the disease of interest: relationship between the Mediterranean diet and antioxidant foods; association of probiotics with the Mediterranean diet; the introduction of the effectiveness of pharmacology.

Bibliography:

They should review the format recommended in this journal for references, there are mistakes in the authors and sometimes it varies if the name of the journal is continued with a comma or not.

Author Response

Referee #2

Dear Referee,

thank you for your comments finalized to improve the submitted paper. The changes are marked in red in the text.

In particular:

- the title has been revised

- the aims of the review are been revised in the introduction section

- Introduction section has been revised and a more complete definition of MD has been reported. However on the basis of current literature, we consider MD not alone, but as a part of an healthy lifestyle, that include the use of probiotics and antioxidants. In this way we describe this association and its synergic effects

- the text has been revised and repetitive sentences are modified

- the suggested references, and in particular international guidelines, are now discussed and reported in the text

- the influence of MD on clinical and biochemical parameters has been discussed

- the role of MD to support standardized treatment for metabolic feature associated with NAFLD has been reported. However, actually a real discussion of the beneficial effects of the diet compared to drugs is limited for the absence of long term randomized trials on this field.

- References section has been revised
